# The shield of self-compassion: A buffer against disordered eating risk from physical appearance perfectionism

Luisa Bergunde, Barbara Dritschel*

School of Psychology and Neuroscience, University of St Andrews, St Andrews, Scotland, United Kingdom

* bd9@st-andrews.ac.uk

**Data Availability Statement:** The research data underpinning this publication can be accessed at https://doi.org/10.17630/39426414-eb9b-4b7f-b078-2f88f96cdd76.

## Abstract

General perfectionistic tendencies as well as perfectionism focussed specifically on one's physical appearance have been implicated as risk factors for disordered eating. This study extends previous research on protective factors by investigating self-compassion as a moderator of the relationship between both general and physical-appearance-perfectionism and disordered eating. A cross-sectional online survey assessed general perfectionism, physical-appearance-perfectionism, disordered eating symptoms, self-compassion and negative affect in female UK university students (N = 421). Results showed physical-appearance-perfectionism explained variance (15%) in disordered eating symptoms above general perfectionism and negative affect. Both perfectionistic concerns about and strivings for appearance perfection were significant unique predictors of disordered eating. Self-compassion moderated the relationship between both perfectionistic concerns and strivings of physical-appearance-perfectionism, but not general perfectionism, and disordered eating. This study suggests both perfectionistic concerns about and strivings for appearance perfection represent potential risk factors for disordered eating among female university students and that self-compassion may reduce their impact.

## Introduction

Disordered eating symptoms, such as dieting, binge eating and unhealthy weight control practices, are highly prevalent in the general population with 6.4% [1] and university students with 20.4% [2]. Engaging in disordered eating increases risk for problematic outcomes, such as full-blown eating disorders [3], obesity [4] and psychological distress [5]. Evidence suggests perfectionism is an important contributor to the development and maintenance of disordered eating symptoms [6], with perfectionism regarding one's physical appearance potentially being particularly salient [7].

### General perfectionism as a risk factor for disordered eating

Perfectionism is a general personality disposition characterized by striving for perfection and setting high personal standards as well as by being overly critical and fearing negative

**Funding:** The authors received no specific funding for this work.

**Competing interests:** The authors have declared that no competing interests exist.

evaluations [8]. It is considered a multidimensional construct [9]. One of the most prominent models of multidimensional perfectionism emphasises both intrapersonal and interpersonal components of perfectionism [10]. This model suggests high standards of perfectionists may be experienced as self-imposed (self-oriented perfectionism) or as imposed by others (socially-prescribed perfectionism). Both self-oriented perfectionism, which involves having disproportionately high personal standards and a motivation to achieve them, and socially-prescribed perfectionism, characterised by beliefs that others hold excessively high standards for oneself, have been linked to disordered eating symptoms, namely dieting and bulimic symptoms respectively, in a non-clinical sample [11].

Other research has demonstrated two broad dimensions of perfectionism, namely perfectionistic concerns and perfectionistic strivings [12], which map onto these two components. Perfectionistic concerns captures worries associated with perfectionism, includes socially-prescribed-perfectionism as well as other perfectionism components, such as concern over mistakes, and has been associated with psychopathology. In contrast, perfectionistic strivings captures high standards and striving, includes self-oriented-perfectionism as well as other perfectionism components, such as personal standards, and has been linked to wellbeing [10, 13]. However, perfectionistic strivings can also become unhealthy and result in clinically-relevant perfectionism when perfectionistic strivings are over-valued and all-consuming [14].

With regards to eating pathology, research has found evidence suggesting both perfectionistic concerns and perfectionistic strivings to be uniquely related to increased eating pathology in non-clinical and clinical populations both cross-sectionally and longitudinally (see [15] for a review). Recent experimental studies further support both perfectionistic concerns and strivings perfectionism as potential causal risk factors of disordered eating symptoms [16, 17]. Findings showed that participants who received an experimental induction of perfectionistic concerns or perfectionistic strivings reported higher disordered eating symptoms 24 hours after the manipulation than those who received no induction [17]. It may be that the induction of perfectionistic concerns and perfectionistic strivings lead to the use of maladaptive eating as an emotion regulation coping strategy for negative emotions resulting from the induction. Individuals high in trait general perfectionism frequently experience feelings of incompetence and failure due to a tendency to set, pursue and fail to achieve unattainable goals and perceived expectations [18]. Furthermore, perfectionists frequently employ emotion regulation strategies such as rumination and avoidance in response to failure, which are considered maladaptive in this context [19]. These responses may increase the likelihood of engaging in compensatory behaviours [20]. This could involve losing control by for instance binge eating to avoid negative feelings [21] or an attempt to restore control of one's goals by restricting food intake [20]. This is supported by Rivière and Douilliez' [22] finding that use of these emotion regulation strategies partially explains the relationship between general perfectionism and disordered eating symptoms in female participants.

## Physical-appearance-perfectionism and its relation to eating pathology

Research suggests perfectionists have particularly high standards in highly valued life domains [23]. One such domain may be physical appearance. Qualitative interviews with eating disorder patients showed that in this group, perfectionistic tendencies were often directed towards one's physical appearance [24]. In addition, positive correlations have been found between a measure assessing physical-appearance perfectionism, the physical appearance perfectionism scale (PAPS), and self-reported weight control behaviours [25]. Furthermore, a cross-sectional survey study with university students demonstrated that physical-appearance-perfectionism explained variance in disordered eating symptoms above general perfectionism [7]. These

preliminary findings suggest physical-appearance-perfectionism may represent a salient risk factor for disordered eating symptoms. However, research has not directly tested whether physical-appearance-perfectionism encompasses all the risk for disordered eating symptoms associated with perfectionism. Since other domain-specific perfectionism measures have been found to do so regarding domain-related outcomes [26], it is pertinent to evaluate if general perfectionism still explains variance in disordered eating symptoms above physical-appearance-perfectionism.

Similar to general perfectionism, the PAPS assesses two components of physical appearance perfectionism, worries-about-appearance-imperfections and hope-for-appearance-perfection, intended to represent the general perfectionism subcomponents of perfectionistic concerns and perfectionistic strivings respectively [7]. Research has supported this conceptualization by demonstrating that worries-about-appearance-imperfection correlated more strongly with perfectionistic concerns than perfectionistic strivings of general perfectionism. However, contrary to expectations, hope-for-appearance-perfectionism has not shown a consistently stronger relationship to perfectionistic strivings than perfectionistic concerns [7]. Thus, validity of this subscale may be questionable.

To hope for something signifies wanting something to happen, whereas to strive implies actively making great efforts to achieve something [27]. These definitions highlight that "*hope*" may not adequately capture the active component of perfectionistic strivings which is key to distinguishing perfectionistic tendencies from general wishes [8]. Moreover, hope-for-appearance-perfection has consistently been shown to be either unrelated to or a non-significant predictor of eating pathology [7, 25], despite perfectionistic strivings of general perfectionism conferring unique risk in eating pathology [11]. This highlights a potential pitfall of hope-for-appearance-perfection's miswording, as researchers have concluded perfectionistic strivings regarding physical appearance is unrelated to negative outcomes, such as disordered eating symptoms [7]. To clarify the role of perfectionistic strivings regarding physical appearance in disordered eating symptoms, this study introduced a modified version of the hope-for-appearance-perfection subscale where the word "*hope*" was replaced with "*strive*" with the aim of better reflecting perfectionistic strivings in the domain of physical appearance.

## Self-compassion as a protective factor against disordered eating symptoms

With regards to disordered eating, a recent review concluded self-compassion, a healthy way of relating to one's self in distress [28], is negatively related to eating pathology [29]. In a seminal experimental study of female participants high in restrictive eating, Adams and Leary [30] demonstrated that after consumption of a doughnut, individuals who received a self-compassion induction experienced fewer negative emotions and less disinhibited eating afterwards than those who received no induction.

Self-compassion encompasses three aspects: 1. responding to one's suffering with self-kindness rather than self-judgment, 2. being mindfully aware of rather than overidentifying with one's feelings and thoughts of suffering, and 3. seeing one's suffering as part of a common human experience rather than feeling isolated [31]. In their review, Inwood and Ferrari [32] found self-compassion to be related to mental health partially through emotion regulation, suggesting self-compassion may act as a facilitator of emotion regulation. Additional studies have reported a moderate positive relationship between self-compassion and psychological flexibility, the ability to be consciously present and adapt one's behaviour to the current situation [33, 34]. It may therefore be that self-compassion exerts its protective effect through increased psychological and emotional flexibility, such that context-appropriate responses and emotion regulation strategies are more easily accessible [35, 36]. Findings examining

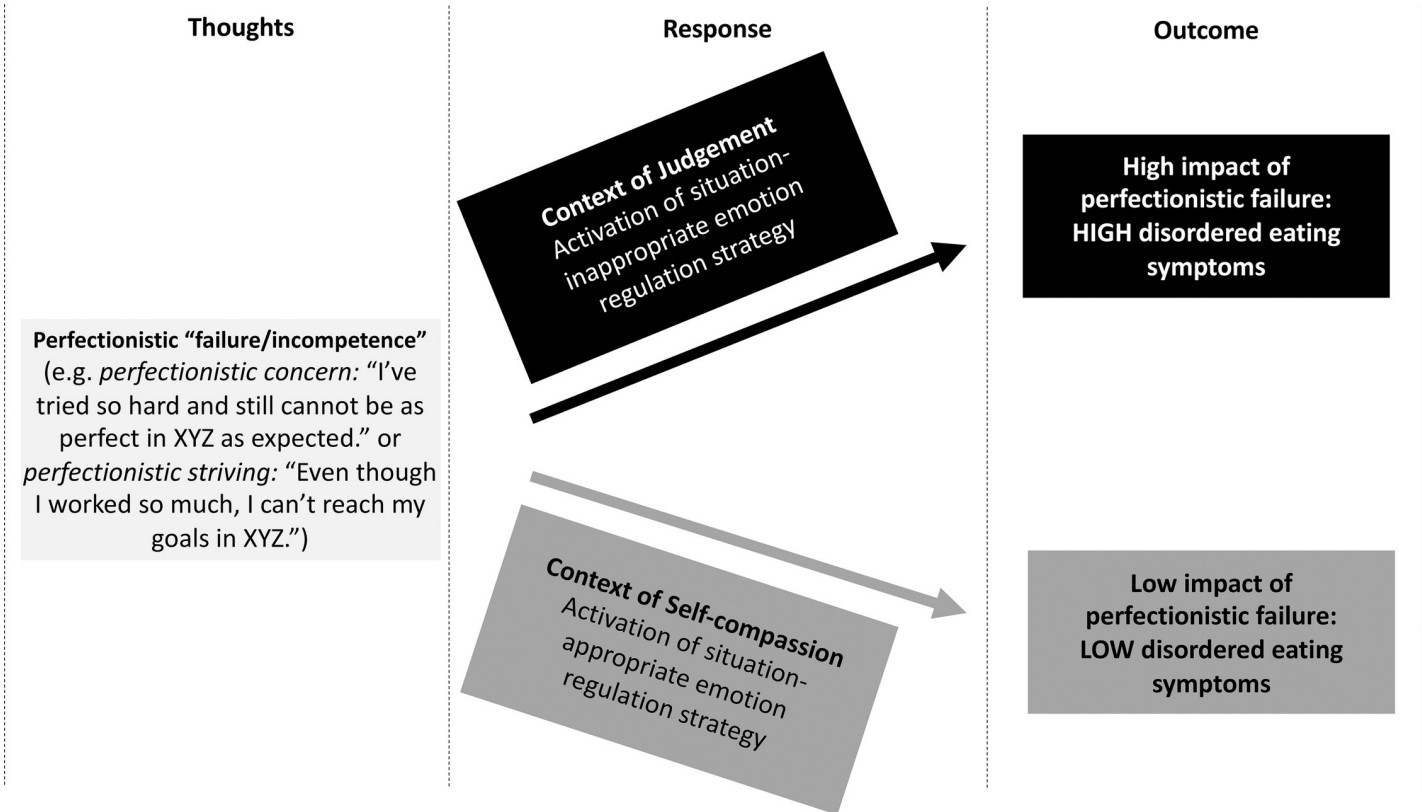

**Fig 1. Theoretical model of the moderating role of self-compassion on the relationship between perfectionism and disordered eating.** XYZ refers to a domain in which an individual is highly perfectionistic.

physiological measures corroborate this by demonstrating that highly self-compassionate individuals exhibit higher resting levels of vagally mediated heart rate variability, an indicator of flexible emotion regulation capacity [37]. This implies self-compassion may enable better adaptation of emotional responses to environmental demands, both physiologically and psychologically. With regards to disordered eating symptoms, self-compassion may be particularly important since emotion regulation difficulties have been found to be associated with disordered eating symptoms [38].

Based on the above, this study proposes a model for self-compassion as a protective factor against perfectionism's effect on disordered eating symptoms (Fig 1). As outlined earlier, perfectionists may be prone to experiences of incompetence [18], which in turn may constitute a situation of suffering [28]. How individuals cope with feelings and thoughts related to their suffering may depend on individual differences in self-compassion. If low on self-compassion, one may be more likely to respond in a context of judgement, thereby potentially limiting the ability to respond with situation-appropriate coping strategies and consequently increasing the likelihood of employing compensatory behaviours, including disordered eating symptoms [21]. In contrast, if high on trait self-compassion, thoughts can be responded to in a context of self-compassion, such that one treats oneself with kindness and sees one's suffering as part of common humanity. This may enable a more flexible and situation-appropriate choice of emotion regulation strategy, thereby reducing the impact of perfectionistic failure on disordered eating symptoms [32].

## Present study

This study aimed to increase understanding of physical-appearance-perfectionism facets as risk factors of disordered eating symptoms and investigate trait self-compassion as a protective factor against the impact of perfectionism, both general and physical-appearance-perfectionism, on eating pathology. Given challenges in the conceptualization and valid assessment of disordered eating symptoms in males, this study limited recruitment to female university students [2, 39]. BMI and negative affect were included in our model as both have demonstrated significant associations with disordered eating symptoms and perfectionism's unique relationship with disordered eating symptoms above these variables remains to be investigated [40].

General perfectionism was expected to explain variance in disordered eating symptoms beyond physical-appearance-perfectionism. We also hypothesised that physical-appearance-perfectionism would predict variance in disordered eating symptoms above perfectionistic concerns and perfectionistic strivings of general perfectionism and negative affect and that both the perfectionistic concerns and a modified perfectionistic strivings subscale of physical-appearance-perfectionism would predict unique variance in disordered eating symptoms. Finally, this study investigated whether the relationship between perfectionism facets and disordered eating symptoms would be moderated by the level of trait self-compassion. Negative affect was included as a covariate to control for the effect of differences in current mood on perfectionism and disordered eating symptoms [41]. The central hypothesis was that the relationship between perfectionism facets and disordered eating symptoms would be weaker the higher the level of trait self-compassion regardless of current negative affect.

## Methods

### Participants

From 691 people who opened the survey, 270 participants were excluded from further analysis. Inclusion criteria were: aged 18 years or over, female and a university student. To ensure high data quality [42], exclusion criteria included: completion of survey in less than 60% of projected time ($\leq$9min) and omission of more than 20% of questionnaire questions. The final sample comprised 421 female university students ($M_{age}$ = 20.95; $SD$ = 3.30). Participants self-described their ethnicity as White (86.9%), Asian (6.0%), Mixed (4.0%), Black (1.4%) and Other (1.2%). Participants were recruited using opportunity sampling with the survey being advertised online for 12 weeks.

### Measures

*Body Mass Index* (BMI) calculated from self-reported height and weight ($kg/m^2$) was normal (18.5$\geq$BMI$\leq$24.99) in 73.5% of students, whereas 9.7% were underweight (BMI<18.5), 13.2% were overweight (25$\geq$BMI$\leq$29.99) and 3.6% were obese (BMI$\geq$30).

The 15-item short form of the 45-item *Multidimensional Perfectionism Scale* (MPS; [10]; short form: [43]) has shown good construct, convergent and discriminant validity [44]). It was used to measure the two components of general perfectionism shown to be related to disordered eating symptoms [15]: self-oriented-perfectionism (5 items) and socially-prescribed-perfectionism (5 items). Items were responded to on a 7-point Likert scale (1 = *disagree* to 7 = *agree*).

The *Physical Appearance Perfectionism Scale* (PAPS; [25]) is a 12-item self-report questionnaire capturing worries-about-appearance-imperfections (7 items) and hope-for-appearance-perfection (5 items). This study used a modified version of the second subscale, replacing the word "*hope*" with "*strive*". As assumptions of exploratory factor analysis (EFA) were met, an

EFA using principal axis factoring with direct quartimin oblimin rotation was conducted on the 12 PAPS items. Two factors had eigenvalues above 1, were positively correlated ($r$ = .41, $p<$.001) and together explained 73.2% of the variance. Factor 1 represented worries-about-appearance-imperfection and factor 2 represented striving-for-appearance-perfection. Items were assessed via a 5-point Likert scale (1 = *strongly disagree* to 5 = *strongly agree*). PAPS has demonstrated good 4-week test-retest reliability and good preliminary construct validity [25].

The 26-item self-report *Eating Attitudes Test* (EAT-26; [3]) measured disordered eating symptoms. The EAT-26 has been shown to be highly reliable and valid [45]. Three subscales are captured: Dieting (13 items), Bulimia and Food Preoccupation (6 items) and Oral Control (7 items). Participants reported the frequency with which behaviours outlined in items were experienced by responding on a 6-point Likert scale (1 = *never* to 6 = *always*). As per Garner et al. [3], 20% of participants were at risk of an eating disorder, reflected by a score of 20 or above.

The 26-item *Self-Compassion-Scale* (SCS; [31]) is a self-report questionnaire which has demonstrated good construct, convergent, discriminant and test-retest validity [31]. It assesses how individuals treat themselves in difficult times and comprises six subscales measuring three facets of self-compassion. (1) Self-kindness versus self-judgement (10 items), (2) Common humanity versus isolation (8 items) and (3) Mindfulness versus over-identification (8 items). Items were rated on a 5-point Likert scale (1 = *almost never* to 5 = *almost always*).

The 21-item *Depression, Anxiety and Stress Scale* (DASS-21; [46]) is a highly reliable and valid measure of the frequency and severity of negative affect over the past week [47]. Although the DASS-21 was conceptualised to respectively assess depression, anxiety and stress, recent research suggests a general negative affect factor explains the majority of the variance in the DASS-21 [48]. Hence this study employed the total DASS-21 score as a measure of negative affect. Items were rated on a 4-point Likert scale (0 = *did not apply to me at all* to 3 = *applied to me very much of the time*).

## Procedure

Ethical approval was granted by St Andrews University Research Ethics Committee. The survey was completed online using the platform Qualtrics®. It included demographic questions (age, gender, ethnicity, height, weight, current university student), followed by the questionnaires presented in a randomized order. Participation was rewarded with the chance to win a £20 Amazon voucher.

## Data screening and analysis

As missing data accounted for only 0.25% of all values, missing items were replaced with the item mean [49]. BMI and disordered eating symptoms were strongly positively skewed, likely due to the non-clinical sample. Further, self-oriented perfectionism showed strong negative skew. Inspection of residual scatterplots indicated heteroscedasticity. To adjust for these violations in assumptions, BMI was transformed logarithmically, and square-root-transformations were applied to self-oriented-perfectionism and EAT26 scores (see S1 Table). After applying these transformations, assumptions for regression analysis were better met. Analyses with transformed data showed no differences in significance of results with the exception of one finding. Self-compassion no longer moderated the relationship between socially-prescribed perfectionism and disordered eating symptoms when transformed data was used for analysis. Therefore, p-, t- and F-values for this relationship are reported from analysis using transformed data. All other results are reported from analyses using untransformed data.

A cross-sectional correlational design was used. Data were analysed using SPSS 24 [50] and the PROCESS macros [51].

**Table 1. Summary of descriptive statistics.** This includes mean (M), standard deviation (SD), Cronbach's Alpha ($\alpha$) and bivariate correlations of study variables (N = 421).

| Variable | M | SD | 1 | 2 | 3 | 4 | 5 | 6 | 7 | 8 | $\alpha$ |
|---|---|---|---|---|---|---|---|---|---|---|---|
| 1. Self-oriented perfectionism (MPS) | 26.8 | 6.31 | - | .45* | .30* | .21* | .28* | -.31* | -.08 | .26* | .87 |
| 2. Socially-prescribed perfectionism (MPS) | 18.62 | 6.83 | | - | .22* | .34* | .29* | -.41* | -.04 | .39* | .82 |
| 3. Striving for appearance perfection (PAPS) | 3.67 | 1.01 | | | - | .46* | .40* | -.29* | -.02 | .20* | .91 |
| 4. Worries about appearance imperfection (PAPS) | 3.12 | 1.07 | | | | - | .57* | -.60* | .07 | .49* | .93 |
| 5. Disordered eating symptoms (EAT-26) | 11.45 | 11.74 | | | | | - | -.41* | -.08 | .48* | .91 |
| 6. Self-compassion (SCS) | 2.67 | .70 | | | | | | - | .04 | -.60* | .93 |
| 7. Body Mass Index (BMI) | 22.24 | 3.20 | | | | | | | - | -.06 | - |
| 8. Negative affect (DASS-21) | 20.44 | 13.06 | | | | | | | | - | .93 |

* *Bonferroni-corrected alpha levels were applied (p < .0017; 2-tailed).* MPS (Multidimensional Perfectionism Scale); PAPS (Physical Appearance Perfectionism Scale), EAT-26 (26-item Eating Attitudes Test); SCS (Self-compassion Scale); BMI (Body Mass Index); DASS-21 (21-item Depression, Anxiety and Stress Scale)

## Results

### Preliminary analysis

Table 1 shows descriptive statistics and bivariate correlations. Internal reliability was good to excellent for all variables. Self-compassion correlated negatively with all perfectionism components, disordered eating symptoms and negative affect. Negative affect and all perfectionism components were positively correlated with disordered eating symptoms. Striving-for-appearance-perfection correlated more strongly with self-oriented-perfectionism ($r(421) = .30$, $p<.001$), than socially-prescribed-perfectionism ($r(421) = .22, p<.001$). In contrast, worries-about-appearance-imperfection correlated more strongly with socially-prescribed-perfection-ism ($r(421) = .34, p<.001$) than self-oriented-perfectionism ($r(421) = .21, p<.001$). As BMI was not significantly related to any other variable in this study, it was not included in subsequent analyses.

### Perfectionism and disordered eating symptoms

A two-stage hierarchical regression in which physical-appearance-perfectionism facets was entered in Step 1 and general perfectionism facets in Step 2 significantly predicted variance in disordered eating symptoms ($R^2 = .37$, $F(4, 416) = 60.97$, $p < .001$; adjusted $R^2 = .36$) (see Table 2). General perfectionism accounted for a small, albeit significant unique amount of variance in disordered eating symptoms ($R^2_{change} = .02$, $\Delta F(2, 416) = 6.57$, $p = .002$) above that accounted for by physical-appearance-perfectionism ($R^2_{change} = .35$, $\Delta F(2, 418) = 112.39$, $p < .001$).

A three-stage hierarchical regression showed that negative affect, general perfectionism and physical-appearance-perfectionism significantly predicted variance in disordered eating symptoms ($R^2 = .41$, $F(5, 415) = 57.93$, $p < .001$; adjusted $R^2 = .40$) (see Table 3). Negative affect (Step 1) accounted for 23% of the variance in disordered eating symptoms ($R^2_{change} = .23$, $\Delta F(1, 419) = 123.35$, $p < .001$). Adding general perfectionism (Step 2) explained an additional 3% of variance in disordered eating symptoms ($R^2_{change} = .03$, $\Delta F(2, 417) = 8.43$, $p < .001$). Further, adding physical-appearance-perfectionism (Step 3) accounted for an additional 15% of variance in eating pathology, ($R^2_{change} = .15$, $\Delta F(2, 415) = 54.12$, $p < .001$), with both worries-about-appearance-imperfection ($\beta = .36$, $p < .001$) and striving-for-appearance-perfection ($\beta = .15$, $p = .001$) predicting unique variance.

**Table 2. Hierarchical multiple regression predicting disordered eating symptoms, with 95% confidence intervals reported in parenthesis.** Confidence intervals, standard errors and significance levels based on 2000 bootstrap samples (N = 421).

| Variable | $R^2$ | $\Delta R^2$ | B | SE(B) | $\beta$ | $p$ |
|---|---|---|---|---|---|---|
| Step 1 | .34 | .34 | | | | |
| Constant | | | -12.65 (-16.36, -9.17) | 1.86 | | < .001 |
| Worries about appearance imperfection | | | 5.44 (4.38, 6.48) | .49 | .50 | < .001 |
| Striving for appearance perfection | | | 1.94 (.87, 3.00) | .52 | .17 | .001 |
| Step 2 | .36 | .02 | | | | |
| Constant | | | -17.94 (-22.92, -13.55) | 2.35 | | < .001 |
| Worries about appearance imperfection | | | 5.17 (4.12, 6.20) | .50 | .47 | < .001 |
| Striving for appearance perfection | | | 1.52 (.44, 2.58) | .52 | .13 | .010 |
| Socially-prescribed perfectionism | | | .08 (-.08, .24) | .08 | .04 | .320 |
| Self-oriented perfectionism | | | .23 (.07, .39) | .08 | .12 | .007 |

*Note*: $R^2$ = r_squared. $\Delta R^2$ = change in $R^2$. B = unstandardized regression coefficient. SE(B) = standard error B. $\beta$ = standardised regression coefficient.

## Moderation analyses

Moderations were analysed using PROCESS with 2000 bootstrap samples and heteroscedasticity-consistent standard errors. Disordered eating symptoms were the outcome variable. In each model, negative affect was entered as a covariate, one of the four perfectionism components was entered as the independent variable and self-compassion was entered as the moderating variable.

Including the interaction of self-compassion and self-oriented-perfectionism, $\Delta R^2$ = .005, $\Delta F(1,416)$ = 2.650,p = .104, and self-compassion and socially-prescribed-perfectionism, $\Delta R^2$ = .002,$\Delta F(1,416)$ = 1.323,p = .251, did not significantly increase variance accounted for in disordered eating symptoms. This suggests self-compassion did not moderate the relationship between general perfectionism facets and disordered eating symptoms.

However, the interaction between self-compassion and worries-about-appearance-imperfections significantly increased variance accounted for in disordered eating symptoms, $\Delta R^2$ = .03, $\Delta F(1,416)$ = 21.51,$p<$.001. This indicates self-compassion moderated the relationship between worries-about-appearance-imperfections and disordered eating symptoms. Simple slopes analysis (Fig 2A) demonstrated that worries-about-appearance-imperfection was significantly related to eating pathology at one standard deviation below mean self-compassion score ($B$ = 7.06, $t$ = 10.10,$p<$.001,95%$CI$:5.69,8.43) and at mean self-compassion score ($B$ = 5.16,$t$ = 10.30, $p<$.001,95%$CI$:4.18,6.15). At one standard deviation above mean self-compassion score the relationship was weaker ($B$ = 3.27,$t$ = 5.53,$p<$.001,95%$CI$:2.11,4.43), however it first became non-significant at 1.86 standard deviations above the mean self-compassion score.

Inclusion of the interaction of self-compassion and striving-for-appearance-perfection also resulted in a significant increase of variance accounted for in eating pathology, $\Delta R^2$ = .02,$\Delta F(1,416)$ = 7.38,$p$ = .007, indicating self-compassion moderated the relationship between striving-for-appearance-perfection and disordered eating symptoms. Simple slopes analysis (Fig 2B) showed that the relationship between striving-for-appearance-perfection and disordered

**Table 3. Hierarchical multiple regression predicting disordered eating symptoms, with 95% confidence intervals reported in parenthesis.** Confidence intervals, standard errors and significance levels based on 2000 bootstrap samples (N = 421).

| Variable | $R^2$ | $\Delta R^2$ | B | SE(B) | $\beta$ | p |
|---|---|---|---|---|---|---|
| Step 1 | .23 | .23 | | | | |
| Constant | | | 2.68 (1.01, .52) | .94 | | .004 |
| DASS21 | | | .43 (.33, .52) | .04 | .48 | < .001 |
| Step 2 | .26 | .03 | | | | |
| Constant | | | -5.54 (-9.46, -1.66) | 2.21 | | .013 |
| DASS21 | | | .37 (.28, .46) | .04 | .42 | < .001 |
| Socially prescribed perfectionism | | | .10 (-.07, .28) | .09 | .06 | .229 |
| Self-oriented perfectionism | | | .28 (.12, .45) | .09 | .15 | .002 |
| Step 3 | .41 | .15 | | | | |
| Constant | | | -16.77 (-21.57, -12.52) | 2.29 | | < .001 |
| DASS21 | | | .22 (.13, .31) | .04 | .25 | < .001 |
| Socially prescribed perfectionism | | | -.01 (-.18, .14) | .08 | -.01 | .855 |
| Self-oriented perfectionism | | | .19 (.04, .35) | .08 | .10 | .020 |
| Worries about appearance imperfection | | | 3.99 (3.01, 4.95) | .53 | .36 | < .001 |
| Striving for appearance perfection | | | 1.76 (.74, 2.87) | .51 | .15 | .001 |

*Note*: $R^2$ = r_squared. $\Delta R^2$ = change in $R^2$. B = unstandardized regression coefficient. SE(B) = standard error *B*. $\beta$ = standardised regression coefficient.

eating symptoms was significant at one standard deviation below the mean self-compassion score ($B = 5.01, t = 5.42, p < .001, 95\% CI: 3.19, 6.82$) and at the mean self-compassion score ($B = 3.49, t = 6.65, p < .001, 95\% CI: 2.46, 4.52$). Whilst attenuated at one standard deviation above the mean self-compassion score ($B = 1.98, t = 3.49, p < .001, 95\% CI: .86, 3.09$), the relationship first became non-significant at 1.38 standard deviations above the mean self-compassion score.

## Discussion

This study aimed to increase understanding of physical-appearance-perfectionism as a risk factor of disordered eating symptoms and investigate whether self-compassion buffers the relationship between perfectionism, both general and physical-appearance-perfectionism, and eating pathology in female students. Overall, means and standard deviations of study variables as well as direction and strength of correlations reflected those reported in prior studies using university students [7]. The only unexpected finding was that BMI was not significantly positively correlated with disordered eating symptoms, as documented in previous studies [52]. This discrepancy may be due to measurement error as BMI findings are not consistent with population statistics in this age group [53]. Based on previous findings on participants showing an underestimation bias in self-reported BMI, it may be that participants in our study failed to accurately report their current height and weight [54].

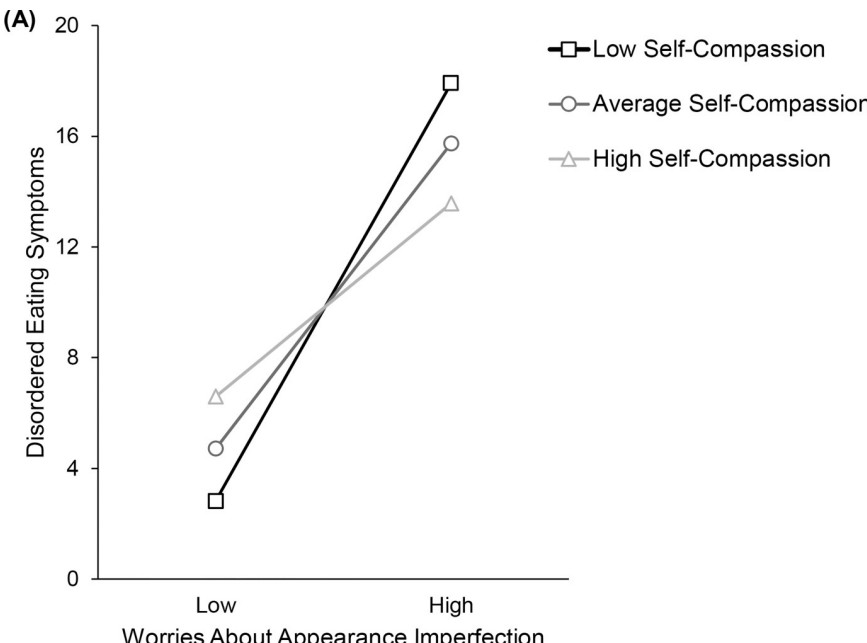

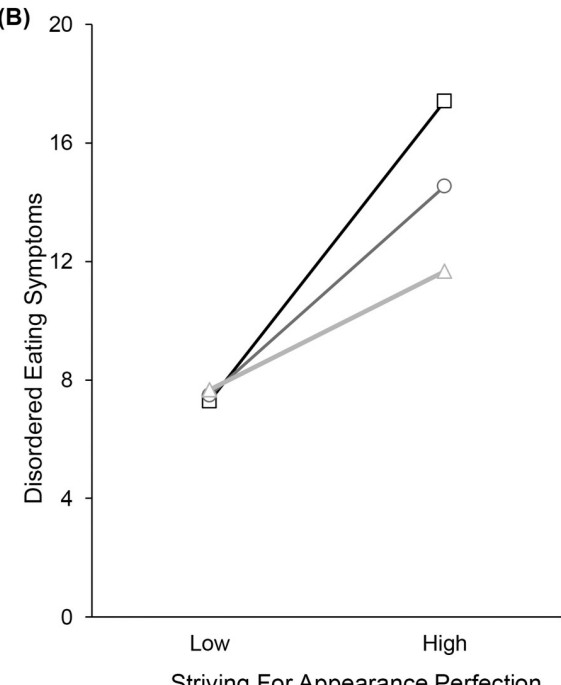

**Fig 2. Regression lines showing the relationship between disordered eating and worries-about- appearance-imperfection (A) and striving-for-appearance-perfection (B) as a function of self-compassion level after controlling for negative affect.** High and low values correspond to ±1SD from the mean.

### Perfectionism is linked to eating pathology

Results showed improved differential construct validity for the modified PAPS. Striving-for-appearance-perfection correlated more strongly with self-oriented perfectionism than socially-prescribed perfectionism and vice versa for worries-about-appearance-imperfections. This

finding contrasts with previous studies which have not found support for this pattern of correlations with the original PAPS, particularly regarding the hope-for-appearance-perfection subscale [7]. However, findings are consistent with research on other multi-dimensional measures of perfectionism showing similar correlation patterns [55, 26]. Consequently, results suggest the modified striving-for-appearance-perfection subscale may better reflect the active striving of perfectionistic strivings than hope-for-appearance-perfection, thereby aligning the PAPS more closely conceptually with the two-dimensional model of perfectionism [13].

The predictive power of physical-appearance-perfectionism was compared to that of general perfectionism in explaining disordered eating symptoms. As expected, results showed that although physical-appearance-perfectionism explained most of the variance in disordered eating symptoms, general perfectionism explained additional unique variance. On the one hand, this contradicts prior research on other domain-specific forms of perfectionism, which fully encompassed the variance explained by general perfectionism in domain-related outcomes [26]. On the other hand, findings are consistent with research showing that being perfectionistic regarding interpersonal relations also increases risk for disordered eating symptoms [56]. Thus, findings indicate being perfectionistic in life domains other than one's physical appearance may also confer unique risk for disordered eating symptoms.

Further regression analyses showed that physical-appearance-perfectionism explained variance in eating pathology above that explained by general perfectionism and negative affect. This replicates prior research and extends it by including negative affect [7]. Moreover, findings contribute to a growing literature demonstrating the added value of domain-specific measures of perfectionism in explaining domain-related outcomes [26, 57]. Findings imply that being perfectionistic regarding one's physical appearance has added explanatory power regarding individual differences in disordered eating symptoms.

Importantly, as expected, both perfectionistic concerns and perfectionistic strivings of physical-appearance-perfectionism emerged as significant predictors of disordered eating symptoms. This contrasts with studies using the original PAPS, which found only worries-about-appearance-imperfection, and not hope-for-appearance-perfection, to uniquely predict eating pathology [7]. Thus, findings suggest actively striving for, as opposed to hoping for, appearance perfection confers unique risk for disordered eating symptoms. This also fits well with the concept of clinical perfectionism [14] as pursuing high standards in the domain of one's physical appearance may constitute an important risk factor for pathological outcomes related to eating. These findings highlight two core points. First, hoping for a perfect appearance may not adequately capture the active striving of perfectionistic strivings perfectionism and future research using the PAPS should deliberate this discrepancy. Second, it is important to consider both perfectionistic strivings and perfectionistic concern regarding physical appearance as potential risk factors for eating pathology in female university students.

## Self-compassion as a moderator

Contrary to hypotheses, strength of the relationship of general perfectionism facets with disordered eating symptoms did not change significantly with self-compassion level, although the non-significant trend was in the direction we predicted, with the perfectionism-disordered eating symptoms relationship being weaker for those high in trait self-compassion. However, consistent with predictions, self-compassion did significantly moderate the relationship between both perfectionistic concerns (worries-about-appearance-imperfections) and perfectionistic strivings (striving-for-appearance-perfection) of physical-appearance-perfectionism and disordered eating symptoms. Generally, these relationships were weaker for women high compared to low in self-compassion. Interestingly, there was a crossover interaction (see

Fig 2A). When worries-about-appearance-imperfections were low those high in self-compassion showed the highest disordered eating. This result may be explained by the effect of a confounding variable not controlled for in this study. For instance, it could be that those individuals reporting low worries-about-appearance-imperfections and high self-compassion were exposed to more weight stigma growing up than the individuals reporting low worries-about-appearance-imperfections and low self-compassion [58]. Overall results indicate that in young women, having a tendency to treat oneself compassionately in times of distress may buffer against the increased disordered eating symptoms associated with higher levels of being perfectionistic regarding one's physical appearance.

Findings corroborate prior research demonstrating negative associations between self-compassion and eating pathology [29]. Moreover, results complement those of Adams and Leary [30] who demonstrated that being more self-compassionate mitigated engagement in disordered eating behaviour. In particular, this study adds to research demonstrating self-compassion as a protective factor against external (media thinness-related pressures [59]), and physical (increased BMI [60]) risk factors of disordered eating symptoms by demonstrating self-compassion may also protect against personality trait risk factors, namely physical-appearance-perfectionism.

The finding that self-compassion only moderated the relation between disordered eating symptoms and physical-appearance-perfectionism, but not general perfectionism, suggests self-compassion may be particularly helpful when dealing with suffering related to one's physical appearance. For instance, self-compassion has shown positive associations with several variables indicative of physical appearance acceptance, such as body image flexibility [60]. Moreover, self-compassion training has been shown to effectively increase body appreciation [61]. It appears that responding kindly and mindfully to worries about or failures in achieving physical-appearance-perfection and viewing associated suffering as common to humanity may facilitate adaptive coping in the eating domain. Consequently, these findings refine our understanding of the relationship between disordered eating, perfectionism and self-compassion as illustrated in the theoretical model in Fig 1. Results suggest self-compassion is a moderator of the relationship between physical-appearance-perfectionism, as opposed to general perfectionism, and disordered eating symptoms. Future research should investigate moderators of the general perfectionism-disordered eating symptoms relationship.

Results indicate that perfectionists who experience frequent feelings of inadequacy regarding physical appearance perfection, but who are able to respond to these in a context of self-compassion, rather than self-judgement, may be able to cope using situation-appropriate emotion regulation strategies, thereby reducing the likelihood of engaging in disordered eating. Consequently, findings indirectly support prior research suggesting self-compassion as a facilitator of appropriate emotion regulation [32], potentially due to its link with psychological flexibility [34]. However, the assumption that a self-compassionate response facilitates more flexible and appropriate emotion regulation remains to be tested directly, for example using a moderated mediation model.

Finally, results provide preliminary evidence that disordered eating prevention programs may benefit from encouraging self-compassion amongst female university students exhibiting high levels of physical-appearance-perfectionism. Importantly, self-compassion has been shown to be both a trait amenable to change and a skill trainable through daily practice and meditation [62], thereby highlighting its utility as an intervention target.

## Limitations and future research

The cross-sectional correlational design impeded ability to determine causality and directionality. Although we suggest that perfectionism leads to disordered eating and this effect may be buffered by self-compassion, this cannot be conclusively determined by our study. Therefore,

future research should employ longitudinal as well as experimental designs to more directly test causality of relationships proposed here and solidify self-compassion as a protective factor. Furthermore, use of self-report questionnaires required self-awareness from participants and entailed increased risk for response bias [63], thereby potentially affecting validity of results. Moreover, since opportunity sampling was employed, the possibility that the sample is not representative of the population exists. It could for example be that the individuals who self-selected to participate shared a particular preoccupation with eating. Also, generalisability of findings is restricted to female university students with a mainly white self-reported ethnicity. It would be interesting to study the proposed relationships in individuals from different cultures, as research suggests cultural variations in trait self-compassion. For instance, individuals from Thailand, where Buddhist teachings are central to the way of life, were found to demonstrate elevated levels of self-compassion compared to individuals from Taiwan and the USA [64]. Research may also benefit from comparing self-compassion's moderating role of the perfectionism-disordered eating symptoms relationship between female and male participants as research suggests trait self-compassion may be higher amongst males [65]. It may also be that elevated trait self-compassion in males may explain why previous studies found no positive association between perfectionism and disordered eating symptoms amongst males [22]. Moreover, this study only examined general disordered eating symptoms, thereby precluding conclusions regarding specific eating disorder symptom groups. Future research should include individual measures of these symptoms to clarify potential differences between symptom groups, as reported elsewhere [11]. Lastly, although a significant moderator, effect sizes were small and self-compassion cannot fully explain existing variance in the relationship between physical-appearance-perfectionism facets and eating pathology. This suggests future research should investigate additional moderators of this relationship and disordered eating symptom prevention efforts should target self-compassion together with other factors.

## Conclusion

This study highlighted the added utility of domain-specific perfectionism measures and suggests both concerns about and striving for a perfect appearance may confer unique risk for disordered eating symptoms in female university students. Moreover, this study provided preliminary evidence that self-compassion may protect against increased disordered eating symptoms associated with increased physical-appearance-perfectionism. Although further clarification of findings is needed, results have the potential to inform and enhance the development of disordered eating prevention. A focus on reducing physical-appearance-perfectionism and developing a more compassionate attitude towards oneself appear crucial.

## Supporting information

**S1 Table. Summary of descriptive statistics for untransformed and transformed values.** This includes mean (M), standard deviation (SD) and range (N = 421).
(PDF)

## Acknowledgments

We thank Mike Oram and Kenneth Mavor for comments on data analysis.

## Author Contributions

**Conceptualization:** Luisa Bergunde.

**Data curation:** Luisa Bergunde.

**Formal analysis:** Luisa Bergunde.

**Methodology:** Luisa Bergunde, Barbara Dritschel.

**Writing – original draft:** Luisa Bergunde.

**Writing – review & editing:** Luisa Bergunde, Barbara Dritschel.

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
