## [Decision Letter · Decision Letter 0]

22 Oct 2019

PONE-D-19-24566

The shield of self-compassion: a buffer against disordered eating risk from physical appearance perfectionism

PLOS ONE

Dear Dr. Dritschel,

Thank you for submitting your manuscript to PLOS ONE. After careful consideration, we feel that it has merit but does not fully meet PLOS ONE’s publication criteria as it currently stands. Therefore, we invite you to submit a revised version of the manuscript that addresses the points raised during the review process.

We would appreciate receiving your revised manuscript by November 20, 2019 To enhance the reproducibility of your results, we recommend that if applicable you deposit your laboratory protocols in protocols.io, where a protocol can be assigned its own identifier (DOI) such that it can be cited independently in the future. For instructions see: http://journals.plos.org/plosone/s/submission-guidelines#loc-laboratory-protocols

We look forward to receiving your revised manuscript.

Kind regards,

Kristin Vickers, Ph.D.

Academic Editor

PLOS ONE

**Journal Requirement****s**

**Other comments**

Dear Dr. Dritschel,

 Thank you for submitting your research to PLOS ONE. I am happy to report that your manuscript entitled "The shield of self-compassion: a buffer against disordered eating risk from physical appearance perfectionism" has received a Minor Revision decision as described below. First, let me apologize the delay in getting feedback to you; a number of experts who were asked to review were unable to do so. Thank you for your patience.

Below you will find helpful comments from two reviewers, along with my input. Required revisions are those that need to be tended to; recommended revisions are those where authors may choose to address the reviewer’s concern, or may choose to explain why they are not making the suggested change. With your revision, please include an explanation of how reviewer comments were addressed. Overall, I find this to be a well done study that contributes to the literature. It will be useful to many researchers. I look forward to your revision and would ask that the revision be received by November 20, 2019.  

Sincerely,

Kristin Vickers

Kristin Vickers, PhD

Comments from Dr. Vickers

Required Revisions

1)            It is extremely important that the data be available. Despite using different computers and browsers, I was not able to open the data using the link provided. Attempts to do so resulted in an error message saying: DOI Not Found  10.17630/39426414-eb9b-4b7f-b078-2f88f96cdd76  Reviewer 2 also noted this issue. 

2)            Please report the reliability values for the measures as reflected in the current sample (as noted by Reviewer 2).

3)            Please present the DASS-21 as 3 subscales (as noted by Reviewer 2).

4)            Please describe the transformations applied and how the transformations were selected (as noted by both reviewers).

5)            Please clarify whether the authors are referring to the analysis performed by the PROCESS macro (for moderation) as regression. (Reviewer 2 also notes this issue, Results comment 5; it appears that the reviewer thought that the PROCESS macros was not used, given the way in which the analyses were described). Did the authors use the PROCESS macro? If so, please clarify the language. As Hays notes (http://processmacro.org/faq.html) the regression approach may yield the same results as PROCESS but differences are also possible.: “…When the same model is estimated using the same data with the same output options, the results will be the same as what you get with SPSS or SAS's regression procedures.  There are many sources of discrepancies you may notice when discrepancies exist...A common one is requesting heteroscedastity-consistent standard errors in PROCESS, which are different than standard OLS standard errors.  SPSS and SAS won't generate these standard errors, but PROCESS will (as will my RLM and HCREG macros) but only if you ask for them.  When you do, standard errors, t-values, p-values, and confidence intervals are different than what SPSS and SAS's internal regression procedures produce, as they should be.”

6)            Please disregard the comments of Reviewer 2 relating to 25% missing data. This was an error on the reviewer’s part, as only 0.25% of the data are missing. Thus, there is no need to address the comments of Reviewer 2 relating to this point, specifically number 3 and number 4 under Methods. Further, it appears the discrepancy between reviewers (Minor versus Major Revision) is a result of this error. With this in mind, and given my own read of this interesting manuscript, I am assigning a Minor Revision decision. 

7)            All other reviewer comments are recommended revisions, and the authors are invited to either address the concerns or explain their rationale for not doing so.

Reviewers' comments:

Reviewer's Responses to Questions

**Comments to the Author**

1. Is the manuscript technically sound, and do the data support the conclusions?

Reviewer #1: Yes

Reviewer #2: Partly

2. Has the statistical analysis been performed appropriately and rigorously? 

Reviewer #1: Yes

Reviewer #2: No

3. Have the authors made all data underlying the findings in their manuscript fully available?

Reviewer #1: Yes

Reviewer #2: No

4. Is the manuscript presented in an intelligible fashion and written in standard English?

Reviewer #1: Yes

Reviewer #2: Yes

5. Review Comments to the Author

Reviewer #1: Comment: Negative Affect (NA) is sometimes presented as a control variable, sometimes as a moderator and sometimes as a predictor. In my experience, it’s customary to make one such prediction rather than casting one variable in all three roles. See the following instances, with line numbers embedded:

Abstract: Results showed physical 33 appearance-perfectionism explained variance (15%) in disordered eating symptoms above 34 general perfectionism and negative affect.

BMI and negative affect were control variables as both 151 have demonstrated significant associations with DES (31). “general perfectionism and negative affect155 and both the PC and a modified PS subscale of physical-appearance-perfectionism would 156 predict unique variance in DES.”

Comment: The weight distribution does not seem to match population stats. Probably because it is self-reported. This idea in the discussion, but only briefly. Please enhance discussion. The way it is currently presented it seems like you are saying BMI doesn’t predict correctly when it really may have been measurement error. See quote:

“BMI 240 was unrelated to all variables and hence not included as a control variable subsequently.”

Stats:

P. 10, You briefly refer to data transformations: give specifics.

Reviewer #2: The shield of self-compassion: a buffer against disordered eating risk from physical appearance perfectionism

Thank you for the opportunity to review this manuscript. The authors conducted an interesting study exploring the dimensionality of perfectionism as it relates to self-compassion, and modelled both in the context of disordered eating in a sample of female university students.

Background and introduction:

1. Literature review for perfectionism is, at times, confusing. From the reviewed articles in the introduction, the consensus appears that various assessments tap into varying dimensions of perfectionism. It is unclear which dimensions are of particular interest to the authors. These sections could benefit from some reorganization to highlight either the existing diversity in the understanding of perfectionism dimensionality, or if the literature is well-established in select dimensions, have a more focused discussion surrounding those specific dimensions of interest.

2. I wonder if the authors could clarify the proposed model and role of self-compassion a little more. In text, the model appears as one of moderation. The graphical representation of this model via Fig. 1 appears more like divergent trajectories marked by adaptive vs. maladaptive coping strategies in response to levels of self-compassion (mediated moderation). Perhaps both in-text description and graphical representation could be better harmonized.

3. Further in response to the discussion of adaptive vs. maladaptive coping strategies (in Fig 1), the emotion regulation literature has also suggested that psychological flexibility is more adaptive overall as strategies like reappraisal may not always be used “adaptively”. For example, some may reappraise the situation to be much more negative than it is. In addition, previously regarded negative coping, such as avoidance, has some basis from a functional contextualism view. Self-compassion may be more closely tied to psychological flexibility, which speaks to the underlying mechanisms of self-compassion, including acceptance, mindful awareness, and non-judgement (DOI: 10.1891/0889-8391.30.1.60).

4. Lines 149-150. With changes in self-perception in the age of social media, a challenge may be that the manifestations of DES is not well conceptualized in males. I suggest the authors rephrase this sentence to the tune of… “Given challenges in the conceptualizations and manifestations of DES in males, we limited the recruitment of only female university students…”.

5. Lines 159-159. Since the authors only recruited females in their study, the statement in relation to women does not apply. Suggest to reword.

Methods:

1. Aside from the possibility of receiving a gift card, where participants otherwise compensated for their time? If not, please discuss limitations of opportunity sampling in relation to self-selection bias in the discussion section.

2. Please include reliability statistics of the sampled population (Cronbach’s alpha) for each measure used.

3. 25% missing data is a significant amount to be replaced with mean. Were the mean replacement values the sample mean or individual mean? Were the missing data missing at random, or more concentrated to specific measures?

4. Given the sensitivity of modelling new relationships and the large quantity of missing data, I suggest the authors use an alternative method to deal with missing values that accounts for more error biases in data (e.g., see https://www.ncbi.nlm.nih.gov/pmc/articles/PMC3668100/)

5. What data transformation was used? Would be helpful if authors produced a table (supplementary is also okay) documenting the raw means, sd, range and the transformed values.

Results:

1. Please note that the DASS-21 is a self-report measure of depression, anxiety, and stress, and should be reported as three separate subscales rather than collapsed into a single variable (Lovibond, S.H. & Lovibond, P.F. (1995). Manual for the Depression Anxiety & Stress Scales. (2nd Ed.)Sydney: Psychology Foundation.). Subsequent analyses should adopt the three-subscale variable structure, and re-analyzed.

2. The rationale for why general perfectionism was entered in step 2 of the hierarchical regression (rather than step 1) was not clearly articulated.

3. Would the authors please include a table of model coefficients for all variables in the hierarchical regressions conducted?

4. Did the authors account for the shared variance of these variables prior to the regression through a single block entry? If not, what was the justification for the order of entry given that correlations at least suggest there may be some overlap or shared variance?

5. Line 228 indicated PROCESS macro would be used for the purpose of conducting moderation. Why did the authors choose a hierarchical regression in lieu of PROCESS?

Discussion:

1. A central theme of this paper is the author’s modification of the PAPS from “hope” to “strive”. Could the authors elaborate more on why a change in terminology may have tapped into a different dimension that the previous version was not successful in tapping into?

2. Can the authors speak more to why there was an interaction with worries-about-appearance imperfection when those in low perfectionism and high in self-compassion also were showed highest in disordered eating? What are some of the potential confounds not otherwise controlled for? For example, if it is the existence of high distress, the “stress” subscale of the DASS-21 would at least tap into the self-reported aspect of it (and similarly for self-reported levels of depression and anxiety).

3. Given the interaction that did not support the hypothesized role of self-compassion as a moderator of perfectionism and DES, the extent to which results support self-compassion as a moderator should be more cautiously interpreted.

4. It would be important for the authors to discuss some cultural implications and variations in self-compassion as it relates to disordered eating and coping, as well as discuss how results may or may not be different in a male sample, older/younger samples, etc.

Minor:

• There are several instances where the paragraph breakdown appears misaligned. This is likely an issue when adapting word format to pdf, or vice versa. May be worth fixing for ease of review.

6. PLOS authors have the option to publish the peer review history of their article (what does this mean?). If published, this will include your full peer review and any attached files.

Reviewer #1: Yes: Karen Shackleford

Reviewer #2: No

---

## [Author Response · Author response to Decision Letter 0]

9 Dec 2019

Dear Dr. Vickers, 

Thank you for identifying areas in need of improvement in our paper titled “The shield of self-compassion: a buffer against disordered eating risk from physical appearance perfectionism” and thereby giving us the opportunity to strengthen our research prior to publication. We appreciate the time and effort you and the reviewers have dedicated to delivering detailed feedback on our manuscript. We have been able to incorporate changes to reflect most of the suggestions provided by the reviewers. 

Here is a point-by-point response to the reviewers’ comments and concerns.

Data Availability

Required Revision 1: It is extremely important that the data be available. Despite using different computers and browsers, I was not able to open the data using the link provided. Attempts to do so resulted in an error message saying: DOI Not Found 10.17630/39426414-eb9b-4b7f-b078-2f88f96cdd76 Reviewer 2 also noted this issue. 

Response: Thank you for pointing this out. We have been in contact with the University of St Andrews Research Data and Information Services. The DOI (https://doi.org/10.17630/39426414-eb9b-4b7f-b078-2f88f96cdd76) has now been activated and the data supporting our research can be found here.

Introduction

Reviewer #1

Comment 1: Negative Affect (NA) is sometimes presented as a control variable, sometimes as a moderator and sometimes as a predictor. In my experience, it’s customary to make one such prediction rather than casting one variable in all three roles. See the following instances, with line numbers embedded

Response: Thank you for pointing this out. In the hierarchical regressions we included negative affect as a predictor in order to assess the effects of general and physical appearance perfectionism above this known predictor of disordered eating symptoms. Negative affect was used as a control variable in the moderation analysis as prior research suggests it predicts variance both in perfectionism levels and disordered eating symptoms. We have made this distinction clearer in the introduction and results. 

Reviewer #2

Comment 1: Literature review for perfectionism is, at times, confusing. From the reviewed articles in the introduction, the consensus appears that various assessments tap into varying dimensions of perfectionism. It is unclear which dimensions are of particular interest to the authors. These sections could benefit from some reorganization to highlight either the existing diversity in the understanding of perfectionism dimensionality, or if the literature is well-established in select dimensions, have a more focused discussion surrounding those specific dimensions of interest.

Response: Thank you for highlighting this. We have extended our discussion of the dimensions of perfectionism which are of particular interest in our study and have given more detailed definitions of the respective dimensions in our introduction. As we are examining general perfectionism and physical appearance perfectionism we focus on these constructs. We have expanded our discussion of general perfectionism to indicate why we are focussing on its subcomponents of striving and concern.

Comment 2: I wonder if the authors could clarify the proposed model and role of self-compassion a little more. In text, the model appears as one of moderation. The graphical representation of this model via Fig. 1 appears more like divergent trajectories marked by adaptive vs. maladaptive coping strategies in response to levels of self-compassion (mediated moderation). Perhaps both in-text description and graphical representation could be better harmonized.

Response: Thank you for this helpful comment. We have addressed it by changing the wording in figure 1. The mediating role of emotion regulation is simply a suggestion for a mechanism as to how or why self-compassion may exert its protective effect. However, in our study we were not testing this and instead concentrating on the role of self-compassion as moderator. 

Comment 3: Further in response to the discussion of adaptive vs. maladaptive coping strategies (in Fig 1), the emotion regulation literature has also suggested that psychological flexibility is more adaptive overall as strategies like reappraisal may not always be used “adaptively”. For example, some may reappraise the situation to be much more negative than it is. In addition, previously regarded negative coping, such as avoidance, has some basis from a functional contextualism view. Self-compassion may be more closely tied to psychological flexibility, which speaks to the underlying mechanisms of self-compassion, including acceptance, mindful awareness, and non-judgement (DOI: 10.1891/0889-8391.30.1.60).

Response: Thank you for raising this interesting point. We have included a discussion of the role of psychological flexibility in our introduction and have refrained from labelling emotion regulation strategies as generally adaptive or maladaptive and instead focussed on situation-appropriate strategy selection and use. It is important to point out that although research has found self-compassion and psychological flexibility to be related, self-compassion was found to explain variance in well-being beyond psychological flexibility (Marshall & Brockmann, 2016; DOI: 10.1891/0889-8391.30.1.60).

Comment 4: Lines 149-150. With changes in self-perception in the age of social media, a challenge may be that the manifestations of DES is not well conceptualized in males. I suggest the authors rephrase this sentence to the tune of… “Given challenges in the conceptualizations and manifestations of DES in males, we limited the recruitment of only female university students…”.

Response: Thank you for this helpful comment. We have rephrased the sentence to better justify our choice of sample and have added a recent review paper as a reference, which highlights current lack of valid assessment tools for disordered eating in males. 

Comment 5: Lines 159-159. Since the authors only recruited females in their study, the statement in relation to women does not apply. Suggest to reword.

Response: Thank you. We agree and have reworded the sentence to reflect this comment. 

Methods

Required Revision 1: Please report the reliability values for the measures as reflected in the current sample (as noted by Reviewer 2).

Comment Reviewer #2: Please include reliability statistics of the sampled population (Cronbach’s alpha) for each measure used.

Response: Thank you for this suggestion and highlighting the importance of internal reliability for measures used. However, we would like to point out that Cronbach’s Alpha for each measure used in the current sample is reported in our manuscript in the final column of Table 1. 

Required Revision 2: Please present the DASS-21 as 3 subscales (as noted by Reviewer 2).

Comment Reviewer #2: Please note that the DASS-21 is a self-report measure of depression, anxiety, and stress, and should be reported as three separate subscales rather than collapsed into a single variable (Lovibond, S.H. & Lovibond, P.F. (1995). Manual for the Depression Anxiety & Stress Scales. (2nd Ed.)Sydney: Psychology Foundation.). Subsequent analyses should adopt the three-subscale variable structure, and re-analyzed.

Response: You have raised an important point here. However, based on recent research findings from Osman et al. (2012) - https://doi-org.wwwdb.dbod.de/10.1002/jclp.21908 - we argue that it is more appropriate to represent the DASS21 as a total score of negative affect or general distress. Osman et al. (2012) investigated the factor structure of the DASS21 in a large sample of nonclinical US university students and found strong evidence for a general distress factor which accounted for the majority of the variance in the DASS21 individual items and the total score. They conclude that the DASS21 is “a theoretically relevant measure of negative emotions that include mixed symptoms of depression, anxiety, and stress”. To enhance our manuscript, we have included this reference and an explanation for using the DASS21 as a measure of negative affect in the methods section of our manuscript. 

Required Revision 3: Please describe the transformations applied and how the transformations were selected (as noted by both reviewers).

Comment Reviewer #1: You briefly refer to data transformations: give specifics.

Comment Reviewer #2: What data transformation was used? Would be helpful if authors produced a table (supplementary is also okay) documenting the raw means, sd, range and the transformed values.

Response: We agree and have included a more detailed description of the transformations applied. Following Reviewer 2’s suggestion, we have included a table documenting the raw means, sd, range and transformed values for the respective variables (please see supplementary materials for the table). 

Reviewer #2

Comment 1: Aside from the possibility of receiving a gift card, where participants otherwise compensated for their time? If not, please discuss limitations of opportunity sampling in relation to self-selection bias in the discussion section.

Response: Thank you for this comment. Aside from the possibility of receiving the gift card, participants were not otherwise compensated for their time. Hence, we have now included a discussion of the limitations associated with opportunity sampling. 

Results

Required Revision 1: Please clarify whether the authors are referring to the analysis performed by the PROCESS macro (for moderation) as regression. (Reviewer 2 also notes this issue, Results comment 5; it appears that the reviewer thought that the PROCESS macros was not used, given the way in which the analyses were described). Did the authors use the PROCESS macro? If so, please clarify the language. As Hays notes (http://processmacro.org/faq.html) the regression approach may yield the same results as PROCESS but differences are also possible.: “…When the same model is estimated using the same data with the same output options, the results will be the same as what you get with SPSS or SAS's regression procedures. There are many sources of discrepancies you may notice when discrepancies exist...A common one is requesting heteroscedastity-consistent standard errors in PROCESS, which are different than standard OLS standard errors. SPSS and SAS won't generate these standard errors, but PROCESS will (as will my RLM and HCREG macros) but only if you ask for them. When you do, standard errors, t-values, p-values, and confidence intervals are different than what SPSS and SAS's internal regression procedures produce, as they should be.”

Comment Reviewer #2: Line 228 indicated PROCESS macro would be used for the purpose of conducting moderation. Why did the authors choose a hierarchical regression in lieu of PROCESS?

Response: Thank you for highlighting this very important point. We would like to clarify that PROCESS was used in the moderation analyses. We have adjusted our description of the analyses in the results section to reflect this. We have further used heteroscedasticity-consistent standard errors and adjusted the affected results.

Reviewer #2

Comment 1: The rationale for why general perfectionism was entered in step 2 of the hierarchical regression (rather than step 1) was not clearly articulated.

Response: Thank you for pointing this out. Please see our response to comment 3 below for a detailed explanation. We have more clearly explained the rationale for this choice in our manuscript introduction.

Comment 2: Would the authors please include a table of model coefficients for all variables in the hierarchical regressions conducted?

Response: Thank you for your comment. We have included a table of model coefficients for all variables in the two hierarchical regressions conducted. 

Comment 3: Did the authors account for the shared variance of these variables prior to the regression through a single block entry? If not, what was the justification for the order of entry given that correlations at least suggest there may be some overlap or shared variance?

Response: Thank you. We did not use a single block entry to account for the shared variance prior to the regression due to the fact that we had a theoretical rationale for entering our variables in the order that we did. The single block entry method is often considered controversial (Tabachnik & Fidell, 2007) and mainly useful in exploratory research. The rationale for entering physical appearance perfectionism first and then general perfectionism was to assess the unique amount of variance accounted for in disordered eating symptoms by general perfectionism above that accounted for by physical appearance perfectionism. In the second hierarchical regression, negative affect was entered first to assess the variance uniquely accounted for by this known risk factor. Then general perfectionism was entered to again assess the unique variance accounted for in disordered eating above that accounted for by negative affect. Finally, physical appearance perfectionism was entered, our main variable of interest in this case, to accurately assess the unique additional variance explained in disordered eating by physical appearance perfectionism. Where possible, we have attempted to make this rationale clearer in our manuscript in the introduction. 

Discussion

Reviewer #1

Comment 1: The weight distribution does not seem to match population stats. Probably because it is self-reported. This idea in the discussion, but only briefly. Please enhance discussion. The way it is currently presented it seems like you are saying BMI doesn’t predict correctly when it really may have been measurement error. See quote:“BMI 240 was unrelated to all variables and hence not included as a control variable subsequently.”

Response: Thank you for pointing this out. We have addressed this both in the results and the discussion by explaining in more detail that this finding was unexpected and that it may be due to measurement error. 

Reviewer #2:

Comment 1: A central theme of this paper is the author’s modification of the PAPS from “hope” to “strive”. Could the authors elaborate more on why a change in terminology may have tapped into a different dimension that the previous version was not successful in tapping into?

Response: Thank you for your comment. We have included a more detailed description of the rationale for the modification in the introduction as well as an additional sentence in the discussion. 

Comment 2: Can the authors speak more to why there was an interaction with worries-about-appearance imperfection when those in low perfectionism and high in self-compassion also were showed highest in disordered eating? What are some of the potential confounds not otherwise controlled for? For example, if it is the existence of high distress, the “stress” subscale of the DASS-21 would at least tap into the self-reported aspect of it (and similarly for self-reported levels of depression and anxiety).

Response: Thank you for your comment. We have included your advice by elaborating in the discussion session on potential variables which may explain the crossover interaction but were not included in our study. We already controlled for negative affect in our moderation analyses. We only controlled for negative affect as one variable based on the findings from Osman et al (2012), which suggest that a general distress factor explains more variance in DASS-21 than the separate subscales of depression, anxiety and stress. 

Comment 3: Given the interaction that did not support the hypothesized role of self-compassion as a moderator of perfectionism and DES, the extent to which results support self-compassion as a moderator should be more cautiously interpreted.

Response: Thank you. We see your point and added a cautioning statement in our limitations section. However, considering the large sample and the use of robust methods, we do believe it is valuable to highlight that the findings suggest self-compassion may function as a moderator. 

Comment 4: It would be important for the authors to discuss some cultural implications and variations in self-compassion as it relates to disordered eating and coping, as well as discuss how results may or may not be different in a male sample, older/younger samples, etc.

Response: We agree. To address this comment, we have included a more detailed discussion of our sample in the limitations section of the discussion. 

We look forward to hearing from you in due time regarding our submission and to respond to any further questions and comments you may have. 

Sincerely, 

Barbara Dritschel and Luisa Bergunde

---

## [Editor Report · Decision Letter 1]

23 Dec 2019

The shield of self-compassion: a buffer against disordered eating risk from physical appearance perfectionism

PONE-D-19-24566R1

Dear Dr. Dritschel,

We are pleased to inform you that your manuscript has been judged scientifically suitable for publication and will be formally accepted for publication once it complies with all outstanding technical requirements.

With kind regards,

Kristin Vickers, Ph.D.

Academic Editor

PLOS ONE

Additional Editor Comments:

The authors did a truly outstanding job in addressing each and every comment made by reviewers in a thoughtful and detailed fashion. The authors presented a thorough revision and a much improved manuscript. Care was taken to make many changes to the manuscript, and the authors also advanced well-explained reasoning (backed up by references) for why a few comments were not actioned.

---

## [Editor Report · Acceptance letter]

2 Jan 2020

PONE-D-19-24566R1 

The shield of self-compassion: a buffer against disordered eating risk from physical appearance perfectionism 

Dear Dr. Dritschel:

I am pleased to inform you that your manuscript has been deemed suitable for publication in PLOS ONE. Congratulations! Your manuscript is now with our production department. 

With kind regards,

on behalf of

Dr. Kristin Vickers 

Academic Editor

PLOS ONE